# Leisure Agriculture and Rural Tourism Benefit Analysis on Eco-Environmental Resource Use

**Baoding Sun [1], Guixia Wang [1,*] and Yunjia Liu [2]**

1   College of Economics and Management, Jilin Agricultural University, Changchun 130118, China
2   College of Chinese Medicinal Materials, Jilin Agricultural University, Changchun 130118, China
*   Correspondence: gxiawang@163.com

**Abstract:** In recent years, particularly, the expansion of tourism has become more and more prosperous, and along with it, the impact on the natural environment has become greater and greater. As a result of the continuous development of the economy, human activity is having a greater impact on the natural environment and agricultural depth. The desire to feel more connected to nature and life is leading an increasing number of people to relocate to more rural areas. Because of this, the management and preparation of rural tourism destinations are of utmost significance. This paper presents a study on the analysis of the benefits of the use of ecological and environmental resources driven by the development of leisure agriculture and rural tourism. The study was carried out by the Environmental Economics and Policy Group (EEPG). The primary purpose of this study is to conduct an analysis of the benefits of agroecological environment and resource use in Jilin Province in accordance with the evaluation system of resource and environmental benefits of leisure agriculture. This evaluation will be combined with an analysis of the current situation of resource use and will be based on the principle of net social benefits of resource and environmental supply. The results of the experiments show that the coupling degree of the agricultural economic system in Jilin Province from 2015 to 2021 is greater than 0.9, which places it in the category of high-quality coupling ($0.9 \leq C \leq 1$). The degree of synergy ranges from 0.54608 to 0.78358 and exhibits an upward trend, but it remains in the medium synergy stage ($0.50 \leq D \leq 8$). This paper carries out relevant research on ecological and environmental resource use, which is of great practical significance in promoting the rational use of leisure agricultural resources, and, ultimately, the long-term sustainable development of leisure agriculture. In addition, the paper presents an analysis of the benefits of ecological and environmental resource use promoted by the development of leisure agriculture and rural tourism.

**Keywords:** agroecosystems; ecosystem planning; rural tourism; ecological environment; benefit analysis

## 1. Introduction

The concept of "leisure agriculture" refers to a novel approach to conducting business in rural areas that makes use of the natural and cultural assets of those locations in order to foster tourism, recreational activities, and authentic rural experiences. In recent years, as a new modern commercial form that drives rural development, leisure agriculture and rural tourism have become interdependent and have developed synergistically. As a result, leisure agriculture has gradually become an important direction for the development of rural tourism and an important help for the revitalization of rural areas. Rural areas are actively developing and laying out their tourism resources in response to the proposal of the rural revitalization strategy. They are doing so by relying on their own one-of-a-kind tourism resources, and the current state of rural leisure tourism development is favorable [1]. However, in order to better consolidate the achievements of poverty alleviation and effectively promote the revitalization of rural areas, a "key chess" that must be taken in the next rural development layout is how to deeply cultivate the existing tourism resources, further promote the high-quality development of leisure agriculture and

rural tourism, and explore an effective mode of "interaction between industry and village and integration of agriculture and tourism". After years of investigating the integrated growth of agriculture and tourism, a new highlight and new form of business that is driving rural development has gradually emerged: the deep integration of leisure agriculture and rural tourism [2]. The immaturity of the mode and the imperfection of the mechanism are to blame for the fact that, despite the gratifying achievements that leisure agriculture and rural tourism have made, there are still a great number of development shackles that prevent the leisure agriculture and rural tourism industries from achieving high-quality development. The requirements that people have for leading a spiritual life are gradually becoming more stringent, and there is a growing demand for vacation travel as a result of these developments in society. Because the conventional method of tourism has been unable to satisfy the requirements of customers, it is necessary to develop a greater number of innovative tourism projects. Cooperation that not only fosters the growth of tourism but also contributes to the expansion of the local economy has been established through the creation of a combination of agricultural entertainment and rural tourism. This cooperation was achieved through the development of a combination of agricultural entertainment and rural tourism. The growth of urban agriculture, entertainment, and tourism can only be positively supported through the creation of new products, business plans, and increased production [3].

As of right now, the growth of agricultural entertainment enriches the form of tourism, promotes the transformation and innovation of the rural tourism industry, and becomes an important way on the economic development of rural villages. Because of this, it is essential to develop agricultural entertainment and rural village tourism together in a gradual manner. To begin, in terms of the economic benefits, the combination of the development of leisure agriculture and tourism helps to rationally develop and utilize rural resources, thereby promoting the sustainable development of rural economy and gradually improving the living standards of people who live in rural areas. This is made possible through the combination of the development of leisure agriculture and tourism. During the process of planning, it is possible to prevent the wastage of resources that is caused by repeated construction, and it is also possible to actively bring the ecological and cultural advantages of various villages into play, which will allow for an increase in the number of tourists who visit. In addition, we can develop ecological harvesting agriculture and boost the trading of agricultural and ancillary products, both of which are conducive to adjusting and optimizing the agricultural industrial structure in rural areas. These two strategies are intertwined with one another. More employment opportunities have been brought to the farmers in the surrounding areas, particularly as a result of the common development and progress of leisure agriculture and tourism. The labor force of young people in rural areas does not have to be relocated to urban areas; rather, they can also launch businesses in their home communities. This not only raises the economic income of farmers, but it also reduces the rate at which rural populations are getting older [4]. Second, the growth of leisure agriculture and rural tourism together has the potential to generate more significant societal benefits. It is possible for it to play a positive role in promoting the integration of urban and rural areas if it works to increase the communication density between cities and villages as well as the interaction that takes place between the two types of communities. Villagers are more likely to see a comprehensive improvement in their cultural quality if urban tourists bring the ideas and culture of the city with them when they visit the countryside for sightseeing. This allows the villagers to benefit economically and experience ideological progress, which is more conducive to the urban tourists' goal of improving the quality of the villagers' culture overall. In the context of the Rural Revitalization Strategy, the management subjects of leisure agriculture and rural tourism are typically managed by the village committee or by special institutions; however, there are some tourism projects that are led by tourism enterprises that are managed by the tourism enterprises themselves. Therefore, in order to facilitate the joint development of leisure agriculture and rural tourism, the managers of the two entities in

question need to communicate with one another and work in tandem. On the other hand, during this process, there will be instances of overlapping responsibilities, ambiguity, and inadequate responsibilities, all of which have the potential to have a negative impact on the development of high-quality leisure agriculture and rural tourism [5]. It is possible for a good management system and management mode to both contribute to the long-term and sustainable development of an industry and to play a good role in the process of guiding the development of an industry. If high-quality sustainable growth is something that the leisure agriculture and rural tourism industries want to achieve, then the management systems that govern these industries need to be continuously improved. This is the utilization benefit of ecological environment resources, as shown in Figure 1.

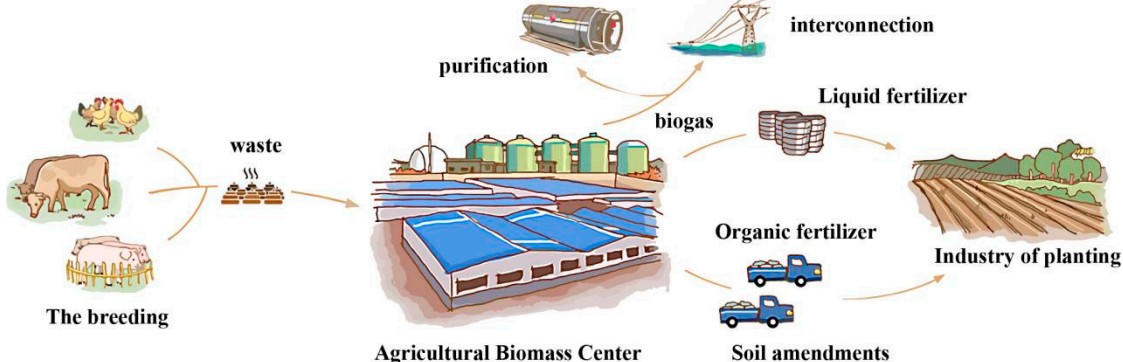

**Figure 1.** Utilization of ecological environment resources.

The development of leisure agriculture and rural tourism is the primary focus of the research and analysis presented in this paper. The paper also studies the utilization benefits of ecological environment resources. The primary purpose of this study is to conduct an analysis of the agricultural ecological environment and the resource utilization benefits in Jilin Province using the evaluation system for leisure agricultural resources and environmental benefits, in conjunction with an analysis of the current situation of resource utilization, and with the principle of resource utilization serving as the foundation. The findings of the experiments indicate that the approach described in this paper is capable of conducting an investigation into the connection that exists between the ecological environment and the utilization of resources in Jilin Province.

## 2. Literature Review

With the improvement of the consumption level and ability of tourism consumers, their requirements for leisure agriculture and rural tourism products have gradually increased. They not only have higher requirements for the quality of products, but also have higher requirements for the application scope and connotation of products [6]. At present, most leisure agriculture and rural tourism products only stay in the surface of rural sightseeing, agricultural product picking and agritainment. Among them, rural scenery sightseeing and agricultural product picking have great seasonal restrictions, and too simple diet, accommodation, and service conditions cannot meet the needs of consumers, and even lead to low consumer satisfaction. The main reason for the superficial development level of tourism products is that the excavation of rural architectural culture, agricultural culture, and other aspects is insufficient [7]. However, if the development level of leisure agriculture and rural tourism products under the Rural Revitalization Strategy is low, the overall economic income of residents will be reduced. Low income will affect the improvement and development of product quality, and ultimately affect the high-quality sustainable development of leisure agriculture and rural tourism. Compared with other occupations, the leisure agriculture and rural tourism industries do not require high skills of the employees, but they need to have certain professional quality and service ability. Moreover, due to the wide coverage of the geographical area, the leisure agriculture and rural tourism require a large

number of employees, which provides many employment opportunities for the surrounding residents [8]. At present, the employees of China's leisure agriculture and rural tourism are mainly local residents, and the overall comprehensive quality and service ability are low. Only by continuously training and introducing professional service personnel can we meet the high-quality development of leisure agriculture and rural tourism under the Rural Revitalization Strategy. In China, the integration of agriculture and rural tourism is still new, and the integration of rural agriculture and rural tourism is unplanned and structured [9]. Although agriculture and rural tourism are well developed in other regions, some regions have failed to follow the agricultural trend of field and city tours. Many farmers have developed agritourism projects, such as farming and fruit picking. Due to the lack of comprehensive planning, investment in support such as tourism is often insufficient. Therefore, agriculture and rural tourism are widespread and unsustainable. Agricultural production and rural tourism not only use the rich resources of rural agricultural areas, but also promote the development of gardening such as roads, food, shelter, business, healthcare, and entertainment [10].

It can be said that the infrastructure plays a larger role in determining whether or not tourists will return than do the agricultural landscape resources themselves. The quality of the infrastructure has a significant bearing on the vacationers' impressions. Although the "last kilometer" of rural development has been open for quite some time, the growth of rural tourism has not been matched by an increase in supporting infrastructure. The critical disaster areas of rural tourism have become issues, such as narrow rural roads, a lack of hotels, unsanitary conditions, and other similar issues [11]. Agricultural ecological environment destruction and agricultural instability have persisted for some time after China's rapid development of its agricultural economy. The agricultural sector and the country's ability to provide food for its people will be severely harmed if this situation persists for an extended period of time in China. Consequently, it is crucial to oversee the environmentally responsible growth of agricultural enterprises. In light of these issues, this article offers an examination and investigation into the promotion of agricultural entertainment, gardening, and urban tourism through the use of ecological environmental resources. China's security and economy rely heavily on the agricultural sector because of the country's status as a major agricultural power. Therefore, how to maximize the benefits of agriculture and tourism by combining the two and how to quantify these gains are important aims. Achieving maximum effectiveness from agricultural practices and supporting development over the long term requires adhering to sound agricultural practices, which allows the growth of a free and sustainable agricultural system [12]. The main points related to the application of the fun farming techniques, along with its hurdles, were uncovered through an examination of the resources related to the development of agricultural production. We propose the necessity of outlining concrete steps for the long-term growth of China's agricultural sector.

## 3. Research Methods

### 3.1. Research on Resource Utilization Benefits

3.1.1. Evaluation System of Leisure Agricultural Resources and Environmental Benefits

The Chinese concept of "leisure agriculture" differs from more conventional farming in that it emphasizes not just the cultivation of crops, but also the cultivation of a wide range of other factors that relate to agriculture. Economic considerations are not the deciding factor in gauging the level of leisure agriculture development. As China's understanding of sustainable development has grown, so too have the methods used to assess the value of agricultural resources used for recreation [13]. The main components of the environmental benefit evaluation system for leisure agricultural resources are the related factors of leisure agricultural resources and the evaluation items of leisure agricultural resources (Table 1). Evaluative factors are those that are expected to have a great deal of bearing on the evaluation itself. The principles of ecological effect, social effect, and humanistic effect guide the selection of relevant factors. Ecological benefit value, social benefit value, and

cultural benefit value are the three primary components of the leisure agriculture resources evaluation project [14]. Ecotourism resources, such as air purification and wound care, are examples of the ecological benefit value of agricultural resources; the social benefit value of agricultural resources, on the other hand, refers to the effects of ecology on society and may be thought of in terms of entertainment value, life comfort value, social harmony value, etc. Agrarian resources have spiritual, cultural, and historical value to humanity (Table 1) [15]. It is clear that the evaluation of shared aspects of resources and the evaluation of shared aspects of resources and environmental benefits are coordinated and mutually supportive of one another. Mastery of the different types of benefits provided by common factors and familiarity with the development policies that correspond to these factors.

**Table 1.** Evaluation system of leisure agricultural resources and environmental benefits.

| Evaluation Items | Evaluation Factor |
| --- | --- |
| Ecological benefit value of leisure agricultural resources | Air purification value; Sewage treatment value; Ornamental value; Rarity value; Natural contribution rate value; Ecological security value |
| Social benefit value of leisure agricultural resources | Entertainment value; The value of life comfort; Value of social harmony |
| Human benefit value of leisure agricultural resources | Historical and cultural values; Spiritual value |

3.1.2. Analysis of Resource Utilization Status

China's share of the world's resources per person is one twelfth of what it is in the United States and one ninth of what it is in the United Kingdom. Yet, because of the continued expansion of China's economy, the demand of Chinese citizens for raw water of a high quality has gradually increased. This is a problem for China's water supply; because of this, there is now a disparity between the needs of Chinese residents and the availability of actual resources [16]. The wide mode of economic development that has been created over a long period of time has led to a poor efficiency of resource consumption. This is the result of the low efficiency of resource usage. This unjustifiable manner of development is also reflected in China's approach to the development and usage of its leisure agricultural resources. The wide style of economic development in China has resulted in a poor level of resource utilization efficiency in rural areas, despite the fact that these rural areas contain a wealth of land and natural resources. As can be observed in Figure 2, the utilization efficiency of leisure agriculture in China is much lower when compared to that of developed countries, such as in Europe and the United States.

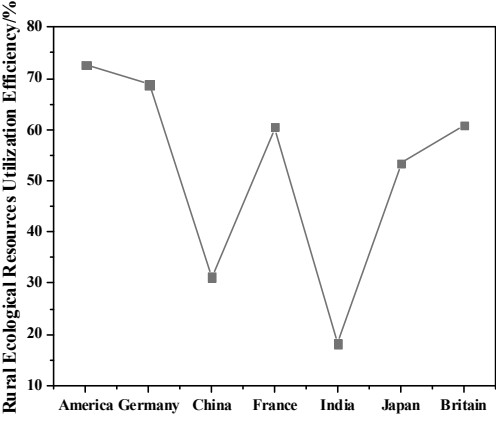

**Figure 2.** Benefits of ecological resources utilization in different countries.

The pursuit of a better quality of life in leisure activities by urban Chinese residents is gradually becoming more successful alongside China's ongoing economic development. When compared to other types of leisure activities, leisure agriculture is distinguished by

a number of distinctive qualities, including a rich rural character and a distinctive rural cultural character. The natural environment is generally superior in rural areas because rural leisure agriculture is concentrated there [17]. It has been discovered, on the other hand, through research and investigation into the utilization mode of leisure agricultural resources in China, that the utilization mode of leisure agricultural resources in China is relatively single, and there is a lack of innovation in the utilization mode of leisure agricultural resources. This was discovered as a result of the investigation and research into the utilization mode of leisure agricultural resources in China. Because of this phenomenon, all of the nation's rural areas now have essentially the same type of tourism for leisure purposes, which lacks any distinguishing features.

### 3.1.3. Social Net Benefit Principle of Resource and Environment Supply

Any scarce good will, in accordance with the supply and demand principle of economics, have both a demand curve and a supply curve, and will arrive at a point of equilibrium at the point where these two curves intersect. This is because any scarce good will have a demand curve that is higher than its supply curve. At this point in time, the entire advantage that society has gained can be defined as the sum total of the consumer surplus as well as the producer surplus (Figure 3), the latter of which is defined by the financial benefit that is obtained by the producer as a result of the creation of the items. This benefit can be described as the difference between the price of the items and the costs that are actually incurred by the producer [18]. Specifically, this benefit is expressed as the profit margin. The term "consumer surplus" refers to the welfare that consumers obtain from the goods, and it is expressed as the difference between the price that consumers are willing to pay for the goods and the price that they actually pay for the goods. Customers are considered to have a "consumer surplus" when the price of the things they purchase falls below their expectations. When environmental resources or environmental quality are viewed as a scarce resource and given market value, environmental resources, similar to other general commodities, have a supply–demand curve and have arrived at a state of equilibrium. This is because environmental resources are similar to general commodities in that they have a market value. This is due to the fact that the environmental resource has been given a market value that has been allocated to it. At this point in time, the social net benefit that is generated by the allocation of environmental resources in the market is equal to the sum of the producer's surplus and the consumer's surplus of the environmental product [19]. This is because the social net benefit that is generated is equal to the sum of the consumer's surplus of the environmental product. The essential point of the principle are as follows: it is possible to use the price that people are willing to pay for environmental resources or services as a measurement of both the level of satisfaction that people have and the value of the environmental products or services that they purchase. This is because people are willing to pay more for higher-quality environmental resources or services. This is true for both items that come from the environment and those that come from resources.

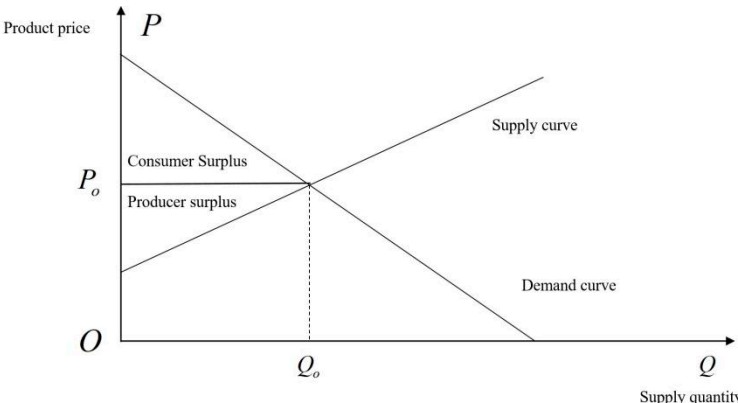

**Figure 3.** Supply–demand curve.

The elimination of pollutants is a primary factor in determining the level of environmental quality or services available in a given area and over a specific period of time [20]. The more pollutants that are eliminated, the higher the quality of the environment will be, and the more environmental services and quality environmental environments will be made available. Therefore, the amount of environmental quality or services provided is equivalent to the elimination of a certain quantity of pollutants, and the amount of environmental quality or services that are still available is equivalent to the number of pollutants that have been eliminated. The total cost and total benefit curve of removing a certain number of pollutants can be obtained from the demand and supply curve for environmental quality and services by replacing the environmental quality or service quantity in the abscissa with the pollutant reduction amount and replacing the product price in the ordinate with the cost or benefit brought by reducing a certain number of pollutants. This will produce the demand and supply curve for the total cost and total benefit of removing a certain number of pollutants. The cost of removing pollutants rises in proportion to the growing quantity of pollutants that need to be eliminated. The more pollutants that are removed, the more quickly the cost will increase; however, the benefits obtained by re-removing pollutants will increase with the increase in the number of pollutants that are removed. The more pollutants that are eliminated, the more gradual the accumulation of benefits will be (see Figure 4).

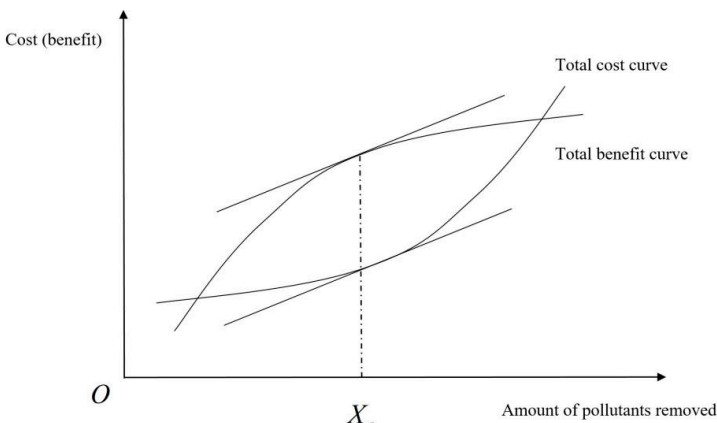

**Figure 4.** Environmental quality and service supply–demand curve.

It is possible to obtain the marginal cost and benefit curve for removing a certain number of pollutants from the total cost and total benefit curve by subtracting the total cost from the total benefit. This will produce the marginal cost and benefit curve. Figure 5 demonstrates that the marginal removal cost increases in proportion to the growing quantity of pollutant that is eliminated, but the marginal removal benefit decreases in proportion to the growing quantity of pollutant that is eradicated [21].

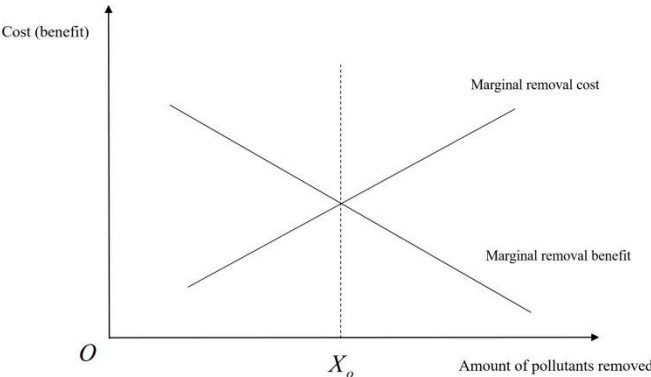

**Figure 5.** Marginal cost and marginal benefit curve of removing a certain number of pollutants.

*3.2. Research on Utilization of Ecological Environment Resources*

3.2.1. Social and Economic Benefits and Ecological Environment Benefits of Land Use

The rate of change in the social and economic benefit system A and the ecological and environmental benefit system B of land use may be described as the following equation; this change is a result of the joint action of their respective index factors and external influence factors:

$$V_A = \frac{\mathrm{d}A}{\mathrm{d}t}, V_B = \frac{\mathrm{d}B}{\mathrm{d}t}, V = f(V_A, V_B) \tag{1}$$

where $V_A$ and $V_B$ represent the evolution speed of system A and system B [22]. Since the whole complex system is composed of two systems, A and B, the evolution speed of the whole complex system can be expressed by $V_A$ and $V_B$ as functions of variables. When studying the whole complex system and the coupling relationship between system A and system B, it can be realized by controlling the two variables $V_A$ and $V_B$. In the coupling model of land use benefit, since the evolution speed $V$ of the whole complex system is affected by two control variables, $V_A$ and $V_B$, they can be used as variables to analyze the evolution type of complex system V in the two-dimensional system ($V_A$, $V_B$) composed of two variables, and the coordinate system with $V_A$ and $V$ as the coordinate system is established. Taking the evolution velocity $V$ of the complex system as the graph of the change trajectory, the included angle between the evolution velocity V of the complex system and the evolution velocity $V$ of ecological and environmental benefits is expressed as $\alpha$, namely, the coupling degree. Then, the coupling degree can be expressed as:

$$\alpha = \arctan V_A / V_B \tag{2}$$

We were able to determine the total value of the social and economic benefits of Changchun City's land use (Figure 6) and the ecological and environmental benefits (Figure 7) that the city will receive from 2007 until 2021 by making use of the formula that was presented earlier in this paragraph. These benefits will accrue to the city over the course of the next decade.

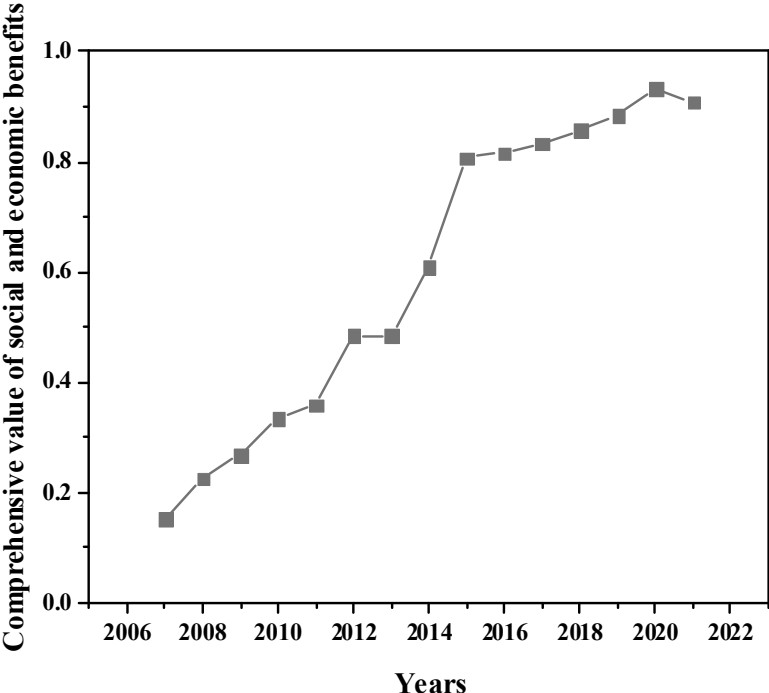

**Figure 6.** Evolution curve of the socioeconomic benefits of land use in Changchun City from 2007 to 2021.

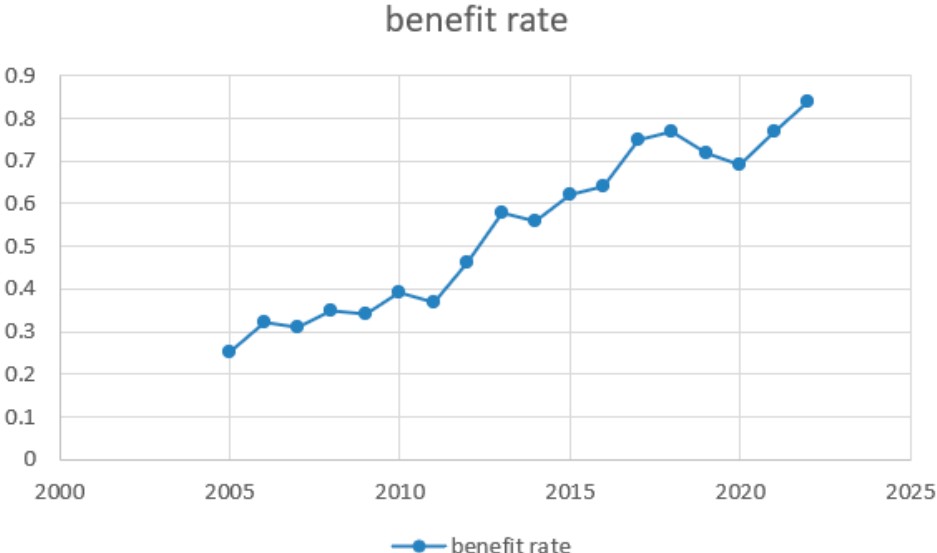

**Figure 7.** Evolution curve of the eco-environmental benefits of land use in Changchun City from 2007 to 2021.

Figures 6 and 7 show that the social and economic benefits of land use in Changchun were generally stable from 2007 to 2021, and increased significantly from 2012 to 2016. This is something that can be seen by looking at both of these figures. This evolutionary trend demonstrates that the economic and social benefits obtained from the research performed on land use in Changchun match the allocation of land resources relatively [23]. This is connected to Changchun's strategy of revitalizing the old industrial base, which the state has been providing strong support for over the past few years. The rapid growth of Changchun's social economy over the past few years demonstrates that the city has the potential to develop into a provincial capital city. The total value of ecological and environmental benefits in Changchun increased from 2007 to 2021, but the trend of evolution was not consistent, and the total value of ecological benefits ranged from 0.25 to 0.83. It is also possible to calculate, in Changchun between the years 2008 and 2021, the degree to which the social and economic benefits of land use are coupled with the ecological and environmental benefits of land use (Figure 8).

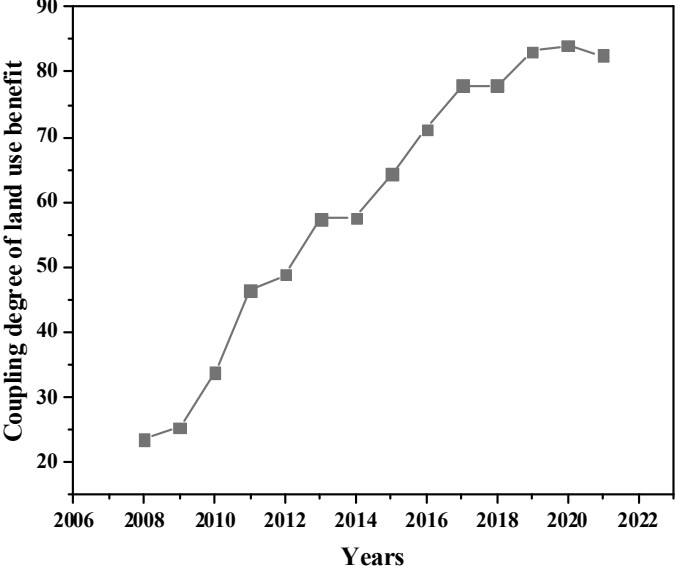

**Figure 8.** Evolution curve of the coupling degree of land use benefits in Changchun City from 2008 to 2021.

3.2.2. Benefit Analysis of the Agricultural Ecological Environment and Resource Utilization in Jilin Province

Jilin had a total population of 26.455 million people as of the end of 2017, with 13.418 million working in agriculture, making up 50.72 percent of the total population. The total area of cultivated land in the province is 6993.38 thousand hectares, which is equivalent to 37.32 percent of the overall land area. In the year 2016, the value of the output generated by aquaculture in Jilin Province accounted for 47.55% of the total value generated by agriculture. The percentage of crop cultivation and harvesting that is done with machinery has reached 86 percent. Rich in rice, corn, and soybeans, the province of Jilin in China is known as the "Golden corn Belt", "hometown of grain and bean", and "hometown of Black land". The output of grain has been steadily increasing, and has now been higher than 50 billion kilograms for four years running, placing it in fourth place in China. The province's average grain yield per unit area has reached 7402.4 kg/ha, which is 260.04 kg/mu higher than the national average of 726.95 kg/mu, representing a 35.8% increase. As a result, the province has ranked first in the nation for several years in a row [24]. According to the existing conditions in Jilin Province, the following 10 factors have been chosen as indicators: per capita arable land area ($hm^2$/person), per capita water resource capacity ($m^3$/person), effective irrigation area (thousand hectares), forest coverage rate (%), fertilizer application rate ($kg/hm^3$), plastic film use rate ($kg/hm^3$), pesticide use rate (kg/ha) $Hm^3$), agricultural disaster area (%), soil and water loss control area (M2), and farmers' per capita disposable income (yuan/year). The first nine indicators have an effect on the agroecological environment system A, which is represented by the letters A1, A2, A3, A4, A5, A6, A7, and A8 in that order. The agricultural economic system B is impacted by the latter seven indicators, which are respectively represented by the numbers B1, B2, B3, B4, B5, B6, and B7. Figure 9 illustrates the system of coupling that exists between the ecological environment of agriculture and the agricultural economy.

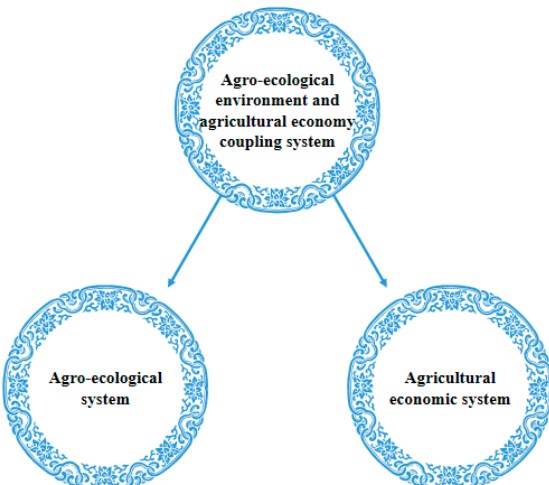

**Figure 9.** Structure diagram of the agricultural coupling system in Jilin Province.

The comprehensive evaluation index model is established. The linear weighted evaluation model is adopted in this study, and the calculation Formula (3) is as follows:

$$U_j = \sum_{i=1}^{n} P_{ij} \cdot w_j \tag{3}$$

When the comprehensive evaluation index is low, the quality of the system is poor and it is unhealthy. The opposite is true when the index is high. If the value is larger, it indicates that the system's quality is higher. The time series line chart of the agricultural ecological environment comprehensive index has been drawn according to the results of the calculations presented earlier, and it can be seen in Figure 10.

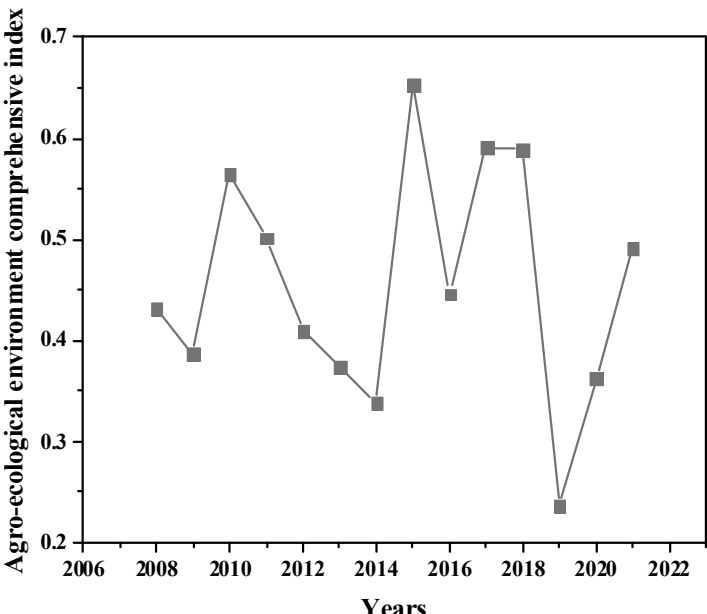

**Figure 10.** Time series broken line of the agroecological environment comprehensive index.

The information that was acquired for the category of general evaluation of the quality of agriculture was as follows, in accordance with the findings of the primary study that was carried out on the evaluation of the quality of agricultural ecological environments. It was determined to provide a natural habitat in the Girin region that is conducive to ecological health (see Table 2).

**Table 2.** Classification benchmark of the comprehensive evaluation index of the agricultural ecological environment quality.

| Composite index | $0 \leq Uj < 0.4$ | $0.4 \leq Uj < 0.6$ | $0.6 \leq Uj < 0.8$ | $0.8 \leq Uj \leq 1$ |
|---|---|---|---|---|
| Quality grade | Poor | Middle | Good | Excellent |

It was observed, through the process of drawing the time series line chart of Jilin Province's agricultural economic comprehensive index (see Figure 11), that the agricultural economy as a whole showed a growing trend, which was in line with the economic development situation. This was because the agricultural economy was in line with the economic development situation. Natural disasters caused a drop in the amount of grain produced per capita in both 2015 and 2017, which led to a decrease in the value-added index of the agricultural industry and a downward trend in the comprehensive evaluation index. If we use 2017 as a cutoff point, we can say that between 2011 and 2017, the growth of the agricultural economy was sluggish, and the index fell as a result of natural disasters in some years. However, the comprehensive evaluation index remained in the range of 0.20584 to 0.39689 throughout this time period. The agricultural economy expanded at a rapid rate from 2017 to 2021, which resulted in significant increases in both the per capita disposable income of farmers and the per capita value of their agricultural output. The comprehensive evaluation index increased from 0.30675 to 0.76233, which corresponds to a total growth rate of 150% as well as an annual growth rate of 16.39%.

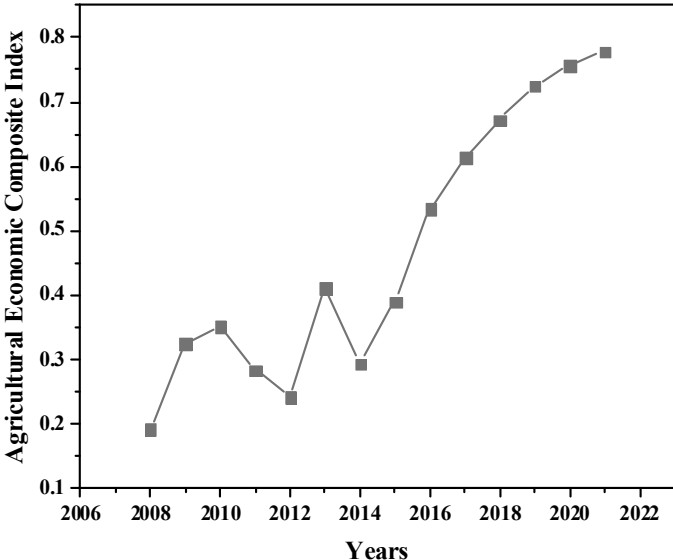

**Figure 11.** Time series line chart of the agricultural economic composite index.

When Figures 10 and 11 are combined into a single figure, as shown in Figure 12, it is possible to observe that the composite index of agricultural economy was lower than the index of agroecological environment prior to 2011, but this disparity began to reverse itself after that year. This is shown by the fact that the index of agricultural economy is now higher than the index of agroecological environment. According to the findings of the relevant research, it can be concluded that Jilin Province belonged to the economic lag type before the year 2011, and that it belonged to the environmental lag type after that year. This conclusion was reached by comparing the two time periods.

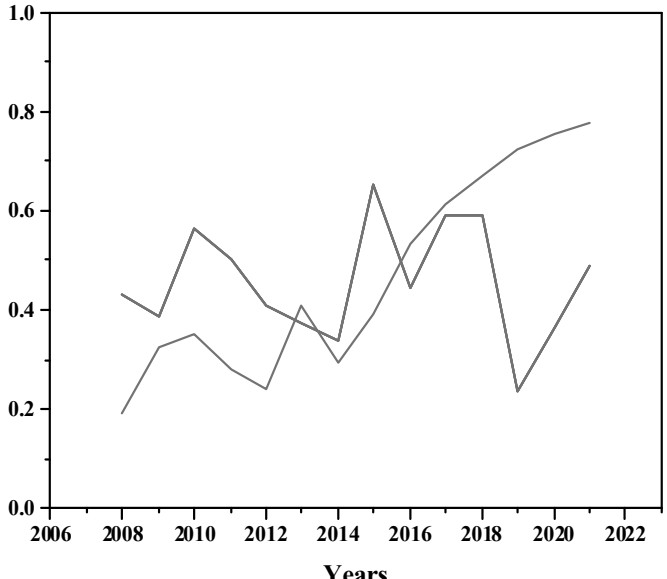

**Figure 12.** Time series of the agricultural ecological environment and the agricultural economy composite index.

### 4. Results of the Analysis

According to a general analysis of the agricultural ecological environment and the agricultural economic environment, the agricultural ecological environment undergoes unpredictable changes, whereas the farm demonstrates consistent expansion and does not demonstrate any connection between the two joint ventures. This finding is based on a comparison of the two environments. Agriculture that is ecologically driven and

agriculture that is economically driven are not dependent on one another but do influence one another [25]. The calculation formula for the interaction and influence between the two systems as well as the calculation formula of the coupling degree and synergy degree of the two systems are shown in Equations (4) and (5), respectively. This will allow for a more accurate analysis of the interaction and influence between the two systems. The level of coupling and synergy between the coupled system of agroecological environment and agricultural economy is outlined in Table 3.

$$C = 2 \times \left[ \frac{Ua \times Ub}{(Ua + Ub)^2} \right]^{\frac{1}{2}} \tag{4}$$

$$D = [C \times (\alpha Ua + \beta Ub)]^{\frac{1}{2}} \tag{5}$$

**Table 3.** Coupling degree and synergy degree of the agroecological environment and the agricultural economy coupling system.

| Year | Degree of Coupling | Degree of Synergy | Year | Degree of Coupling | Degree of Synergy |
|------|--------------------|-------------------|------|--------------------|-------------------|
| 2015 | 0.93502 | 0.54608 | 2010 | 0.9641 | 0.70426 |
| 2016 | 0.97632 | 0.59542 | 2011 | 0.98974 | 0.70986 |
| 2017 | 0.96206 | 0.65688 | 2012 | 0.98956 | 0.76759 |
| 2018 | 0.9612 | 0.6043 | 2013 | 0.98862 | 0.78394 |
| 2019 | 0.96143 | 0.58723 | 2014 | 0.96029 | 0.65364 |
| 2020 | 0.96992 | 0.62593 | 2015 | 0.93227 | 0.72934 |
| 2021 | 0.96826 | 0.57042 | 2016 | 0.97971 | 0.78358 |

Table 3 reveals that the integration of the agricultural ecological environment and the economy of the Girin region is greater than 0.9 from 2015 to 2021, which indicates a good connection ($0.9 \leq C \leq 1$) between the two factors. This can be seen over the course of the table. The degree of synergy can range anywhere from 0.54608 to 0.78358, but it will never be higher than the middle synergy level ($0.50 \leq D < 0.8$). The dotted picture of the agricultural ecological environment and the agricultural economy led to the conclusion that before 2011, the development of the agricultural economy was lagging behind the growth of ecological agriculture, but the relationship between the two is still very strong. This was determined based on the fact that the relationship between the two is still very strong. It is already too late to begin the process of commercializing agriculture, which ultimately results in negative effects on the agricultural ecological environment and the subsequent stages of commercialization of agriculture. The expansion of the agricultural sector brings greater opportunities for cooperation between the two perspectives. On the other hand, the growth of the agricultural industry has not outpaced the growth of the agricultural environment, and this stage is known as a business-as-usual management system. After 2011, the expansion of the agricultural industry caused the development of the agricultural ecological environment to fall further and further behind. Even though the economy is expanding at a rapid rate, the progression of the ecological environment has remained relatively unchanged. When we prioritize economic growth over ecological progress, this demonstrates that we have made a concession in terms of the development of the ecological environment. At this level, the type of business that operates in a regulated setting is described. On the other hand, the fact that the degree of synergy at this stage is growing indicates that the interaction between the two is continually becoming more powerful. Therefore, it is even more necessary to pay attention to ecological environment protection while developing the economy, so that it can help economic development, and allow the two to promote each other and develop together in order to achieve sustainable development. This will ensure that the economy is able to grow in a way that is environmentally friendly.

## 5. Conclusions

The development of leisure agriculture and rural tourism has significant benefits for the ecological environment and resources utilization. Through the promotion of sustainable agriculture practices and the preservation of natural resources, this form of tourism provides an opportunity for both economic development and environmental conservation. One of the primary benefits of leisure agriculture and rural tourism is the conservation and restoration of ecosystems. These tourism activities promote the protection and restoration of natural habitats and wildlife, as well as the sustainable use of natural resources, such as water, soil, and air. This helps to prevent environmental degradation, biodiversity loss, and ecosystem collapse, which are critical issues facing the world today. Another benefit of leisure agriculture and rural tourism is the promotion of sustainable agricultural practices. Through the adoption of ecofriendly farming methods and the encouragement of organic farming, these tourism activities contribute to the reduction of harmful chemicals and pollution in the environment. This results in improved soil quality, cleaner water, and better air quality, which benefits both the environment and human health. Furthermore, the development of leisure agriculture and rural tourism also provides economic benefits to local communities. This form of tourism creates job opportunities, promotes local entrepreneurship, and stimulates economic growth in rural areas. It also contributes to the preservation of cultural heritage and traditional lifestyles, which are important for the conservation of local identities and traditions. In conclusion, the benefits of ecological environment resources utilization driven by the development of leisure agriculture and rural tourism are numerous and significant. These tourism activities promote the conservation and restoration of ecosystems, the adoption of sustainable agricultural practices, and the creation of economic opportunities in rural areas. As such, the promotion and support of leisure agriculture and rural tourism can contribute to a more sustainable and environmentally friendly future.

**Author Contributions:** Conceptualization, B.S.; methodology, G.W.; software, Y.L.; validation, G.W. and Y.L.; project administration, B.S.; funding acquisition, B.S. All authors have read and agreed to the published version of the manuscript.

**Funding:** This study is supported by: (1) The Jilin Provincial Department of Science and Technology Innovation Development Strategy Research Project (20210601001FG): the Promotion of Tourism Accessibility around Changbai Mountain; (2) The Education Department of Jilin Province (JJKH20210386SK): Research on the development effect of leisure agriculture based on farmers' perspective.

**Institutional Review Board Statement:** Not applicable.

**Informed Consent Statement:** Not applicable.

**Data Availability Statement:** The labeled data set used to support the findings of this study is available from the corresponding author upon request.

**Conflicts of Interest:** The author declares that there are no conflict of interest.

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
