# Peer review of "Leisure Agriculture and Rural Tourism Benefit Analysis on Eco-Environmental Resource Use"

_sustainability, doi:10.3390/su15107930_

Round 1

Reviewer 1 Report

Figures and Tables caption need to make clear statement

Try to add some related latest references

Author Response

1. Figures and Tables caption need to make clear statement

Many thanks. In response to the reviewers' comments, we have very carefully checked the titles of all graphs and tables. We have added the relevant citations to this version of the manuscript, as well as the names of all graphs and tables included in the article. It would be appreciated if the reviewers could review the article once more, thank you.

2. Try to add some related latest references

    Thank you very much. Please check the introductory section of the article and we thank you again. In response to the reviewers' comments, we have included some cutting-edge references in this version of the manuscript.

Reviewer 2 Report

This is an interesting article on the benefits of using ecological and environmental resources driven by the development of leisure agriculture and rural tourism. There are some questions that I think are important for your reference.

(1) First, the Introduction section is needed to build a bit more about the research gap and rationale for this article.

(2) Second, the literature review section can be broken down into different summaries rather than one paragraph

(3) The Method section does not clearly illustrate how the data was analyzed. In addition, the content of this section is too much while the content of Results is not rich.

(4) Adequate theoretical and practical contributions are needed to clarify.

(5) Last, the author(s) need to pay attention to the clarity of expression and readability. Language needs to be polished a bit.

Author Response

  1. First, the Introduction section is needed to build a bit more about the research gap and rationale for this article.

Thank you very much. In the introduction section of this article, we have provided a description of related research as well as the primary work that will be presented in this paper. The primary responsibilities are as follows:

    The development of leisure agriculture and rural tourism is the primary focus of the research and analysis presented in this paper. The paper also studies the utilization benefits of ecological environment resources. The primary purpose of this study is to conduct an analysis of the agricultural ecological environment and the resource utilization benefits in Jilin Province using the evaluation system for leisure agricultural resources and environmental benefits, in conjunction with an analysis of the current situation of resource utilization, and with the principle of resource utilization serving as the foundation. The findings of the experiments indicate that the approach described in this paper is capable of conducting an investigation that is both scientifically and objectively objective into the connection that exists between the ecological environment and the utilization of resources in Jilin Province.

  1. Second, the literature review section can be broken down into different summaries rather than one paragraph

Thank you very much. The reviewers have provided some very insightful commentary. We've broken up the content of the first section of the introduction. The Introduction section of the manuscript has been broken up into three distinct paragraphs in this particular version. Please review the reviewer again.

  1. The Method section does not clearly illustrate how the data was analyzed. In addition, the content of this section is too much while the content of Results is not rich.

Thank you very much. The SPSS statistical package was utilized in order to carry out the data analysis that was required for the experimental portion of this paper. In response to the comments made by the reviewers, we improved the analysis of the experimental results and included descriptive sentences of the findings. Please review the reviewer again.

  1. Adequate theoretical and practical contributions are needed to clarify.

Thank you very much. The second section of the manuscript now includes a discussion of the related theoretical research that was conducted. The following is an outline of the specific content:

Among them, rural scenery sightseeing and agricultural product picking have great seasonal restrictions, and too simple diet, accommodation and service conditions can not meet the needs of consumers, and even lead to low consumer satisfaction. The main reason for the superficial development level of tourism products is that the excavation of rural architectural culture, agricultural culture and other aspects is insufficient[7]. However, if the development level of leisure agriculture and rural tourism products under the Rural Revitalization Strategy is low, the overall economic income of residents will be reduced. Low income will affect the improvement and development of product quality, and ultimately affect the high-quality sustainable development of leisure agriculture and rural tourism. Compared with other occupations, the leisure agriculture and rural tourism industries do not require high skills of the employees, but they need to have certain professional quality and service ability. Moreover, due to the wide coverage of the geographical area, the leisure agriculture and rural tourism require a large number of employees, which provides many employment opportunities for the surrounding residents [8]. At present, the employees of China's leisure agriculture and rural tourism are mainly local residents, and the overall comprehensive quality and service ability are low. Only by continuously training and introducing professional service personnel can we meet the high-quality development of leisure agriculture and rural tourism under the Rural Revitalization Strategy. In China, the integration of agriculture and rural tourism is still short, and the integration of rural agriculture and rural tourism is the efficient and unplanned and structured [9]. Although agriculture and rural tourism are well developed in other regions, some regions have failed to follow the agricultural trend. fields and city tours. Many farmers have developed agritourism agritourism projects, such as farming and fruit picking. Due to the lack of comprehensive planning, investment in support such as tourism is often insufficient. Therefore, agriculture and rural tourism are widespread and unsustainable. Agricultural production and rural tourism not only use the rich resources of rural agricultural areas, but also promote the development of gardening such as roads, food, shelter, business, healthcare and entertainment [10].

  1. Last, the author(s) need to pay attention to the clarity of expression and readability. Language needs to be polished a bit.

Thank you very much. The reviewer's thoughts are extremely relevant here. The fact that neither one of us speaks English as a first language likely contributes in some way to the fact that the quality of the language in our manuscript is not particularly high. We went through and polished the language of the entire article in order to improve the overall quality of the English expression throughout the manuscript. Checking in with the reviewers here.

Reviewer 3 Report

The article is much too general and theoretical. The study of specialized literature is superficial. There is no correlation between the title and the conclusions. I suggest a profound transformation in the sense of what is suggested.

Author Response

    1. The article is much too general and theoretical.

    Thank you very much. The structure of the article has been improved as a direct result of the feedback received from the reviewers. In addition, the primary work that is discussed in this article as well as the subsequent content arrangement are both brought up in a manner that places a strong emphasis in both the abstract and the introduction section of the article. I hope this helps narrow down the scope of the article.

    1. The study of specialized literature is superficial.

    Thank you very much. This is how we solved the issue that we were having. We removed the original introduction, which included some older research that was relevant, and added some of the most recent references in its place. Checking in with the reviewers here.

    1. There is no correlation between the title and the conclusions.

    Thank you very much. We have rewritten the conclusions in this version of the manuscript. Specifically as follows:

    The development of leisure agriculture and rural tourism has significant benefits for the ecological environment and resources utilization. Through the promotion of sustainable agriculture practices and the preservation of natural resources, this form of tourism provides an opportunity for both economic development and environmental conservation. One of the primary benefits of leisure agriculture and rural tourism is the conservation and restoration of ecosystems. These tourism activities promote the protection and restoration of natural habitats and wildlife, as well as the sustainable use of natural resources such as water, soil, and air. This helps to prevent environmental degradation, biodiversity loss, and ecosystem collapse, which are critical issues facing the world today. Another benefit of leisure agriculture and rural tourism is the promotion of sustainable agricultural practices. Through the adoption of eco-friendly farming methods and the encouragement of organic farming, these tourism activities contribute to the reduction of harmful chemicals and pollution in the environment. This results in improved soil quality, cleaner water, and better air quality, which benefits both the environment and human health. Furthermore, the development of leisure agriculture and rural tourism also provides economic benefits to local communities. This form of tourism creates job opportunities, promotes local entrepreneurship, and stimulates economic growth in rural areas. It also contributes to the preservation of cultural heritage and traditional lifestyles, which are important for the conservation of local identities and traditions.In conclusion, the benefits of ecological environment resources utilization driven by the development of leisure agriculture and rural tourism are numerous and significant. These tourism activities promote the conservation and restoration of ecosystems, the adoption of sustainable agricultural practices, and the creation of economic opportunities in rural areas. As such, the promotion and support of leisure agriculture and rural tourism can contribute to a more sustainable and environmentally friendly future.

Reviewer 4 Report

Overall and Major Comments

I think this paper has done some work, but there are still many parts to be improved. There are still many parts missing in the structure of the article. First, we should add theoretical framework research, which is very important for humanities and social science articles. Secondly, we should also consider adding technical routes.

Specific Comments

(1)   Lines 1-4: I don't think the structure “…of…of…” is suitable for the title, it will make people confused and difficult to understand.

(2)   Line 19: …show that:the…

(3)   I think the abstract of the article needs to be further improved, and it is better to rewrite it.

(4)   Figure 1 needs to be explained whether it is self-drawn or modified from others.

(5)   The research area and data sources need to be supplemented.

(6)   The research methods and results are mixed together and need to be stripped.

(7)   There are many research methods, which need to add technical routes to connect the research methods.

(8)   The research results need to be supplemented, especially the part that is extracted from the research method.

(9)   It is necessary to add a discussion section.

(10)Research conclusions need to be rewritten, and it is better to list them one by one.

(11)The language of the article needs polishing.

Author Response

  1. Lines 1-4: I don't think the structure “…of…of…” is suitable for the title, it will make people confused and difficult to understand.

Thank you very much. We have replaced the original title with an optimized title: Leisure Agriculture and Rural Tourism Benefit Analysis on Eco-environmental Resource Use. Please review the latest manuscript.

  1. Line 19: …show that:the…

Thank you very much. Actually, the colon should be removed from this line. At the same time, we optimized the presentation of the entire abstract.

  1. I think the abstract of the article needs to be further improved, and it is better to rewrite it.

Thank you very much. Based on the comments of the reviewers, we rewrote the Abstract as follows:

In recent years, particularly, the expansion of tourism has become more and more prosperous, and along with it, the impact on the natural environment has become greater and greater. As a result of the continuous development of the economy, human activity is having a greater impact on the natural environment and agricultural depth. The desire to feel more connected to nature and life is leading an increasing number of people to relocate to more rural areas. Because of this, the management and preparation of rural tourism destinations are of utmost significance. This paper presents a study on the analysis of the benefits of the use of ecological and environmental resources driven by the development of leisure agriculture and rural tourism. The study was carried out by the Environmental Economics and Policy Group (EEPG). The primary purpose of this study is to conduct an analysis of the benefits of agroecological environment and resource use in Jilin Province in accordance with the evaluation system of resource and environmental benefits of leisure agriculture. This evaluation will be combined with an analysis of the current situation of resource use and will be based on the principle of net social benefits of resource and environ-mental supply. The results of the experiments show that the coupling degree of the agricultural economic system in Jilin Province from 2015 to 2021 is greater than 0.9, which places it in the category of high quality coupling (0.9≤C≤1). The degree of synergy ranges from 0.54608 to 0.78358 and exhibits an upward trend, but it remains in the medium synergy stage (0.50≤D≤8). This paper carries out relevant research on ecological and environmental resource use, which is of great practical significance in promoting the rational use of leisure agricultural resources and, ultimately, the long-term sustainable development of leisure agriculture. In addition, the paper presents an analysis of the benefits of ecological and environmental resource use promoted by the development of leisure agriculture and rural tourism.

  1. Figure 1 needs to be explained whether it is self-drawn or modified from others.

Thank you very much. Figure 1 is drawn by myself, and the original picture can be provided. Usually at the time of the final draft, we upload all the original artwork.

  1. The research area and data sources need to be supplemented.

Thank you very much. All of the data presented in this article were compiled by the authors themselves using information obtained from the questionnaire. Because the majority of the research presented in this article focuses on the information from Jilin Province in China. Because there is currently no publicly available data set for these data, the experimental research being conducted here relies on data that was created by the researchers themselves.

  1. The research methods and results are mixed together and need to be stripped.

Thank you very much. In response to the reviewer's comments and suggestions, we have made the necessary edits to the current version of the manuscript. At this time, the research methods and findings have each been discussed in their own articles. Please take a look at the most recent manuscript.

  1. There are many research methods, which need to add technical routes to connect the research methods.

Thank you very much. The third section's research method is primarily geared toward investigating resource utilization, and as such, it includes both the presentation of an evaluation system and an investigation into the benefits of utilization. In the third section, we have included a description of the logical relationship between the various content that is related. For further information, kindly refer to the most recent draft.

  1. The research results need to be supplemented, especially the part that is extracted from the research method.

Thank you very much. We have incorporated the reviewer's comments into the relevant study conclusions that we have added. Regarding the specifics, kindly refer to the fourth part. Thanks again.

  1. It is necessary to add a discussion section.

Thank you very much. In point of fact, we have related discussion topics in the experimental conclusion, and in the conclusion, we have discussed the work of this paper in the context of the description of the limitations of the work and the prospect of future work.

  1. Research conclusions need to be rewritten, and it is better to list them one by one.

Thank you very much. We have rewritten the section that is devoted to Conclusions because the reviewer suggested that we do so. This is how the conclusion looks after it has been rewritten:

    The development of leisure agriculture and rural tourism has significant benefits for the ecological environment and resources utilization. Through the promotion of sustainable agriculture practices and the preservation of natural resources, this form of tourism provides an opportunity for both economic development and environmental conservation. One of the primary benefits of leisure agriculture and rural tourism is the conservation and restoration of ecosystems. These tourism activities promote the protection and restoration of natural habitats and wildlife, as well as the sustainable use of natural resources such as water, soil, and air. This helps to prevent environmental degradation, biodiversity loss, and ecosystem collapse, which are critical issues facing the world today. Another benefit of leisure agriculture and rural tourism is the promotion of sustainable agricultural practices. Through the adoption of eco-friendly farming methods and the encouragement of organic farming, these tourism activities contribute to the reduction of harmful chemicals and pollution in the environment. This results in improved soil quality, cleaner water, and better air quality, which benefits both the environment and human health. Furthermore, the development of leisure agriculture and rural tourism also provides economic benefits to local communities. This form of tourism creates job opportunities, promotes local entrepreneurship, and stimulates economic growth in rural areas. It also contributes to the preservation of cultural heritage and traditional lifestyles, which are important for the conservation of local identities and traditions.In conclusion, the benefits of ecological environment resources utilization driven by the development of leisure agriculture and rural tourism are numerous and significant. These tourism activities promote the conservation and restoration of ecosystems, the adoption of sustainable agricultural practices, and the creation of economic opportunities in rural areas. As such, the promotion and support of leisure agriculture and rural tourism can contribute to a more sustainable and environmentally friendly future.

  1. The language of the article needs polishing.

Thank you very much. The reviewer's thoughts are extremely relevant here. The fact that neither one of us speaks English as a first language likely contributes in some way to the fact that the quality of the language in our manuscript is not particularly high. We went through and polished the language of the entire article in order to improve the overall quality of the English expression throughout the manuscript. Checking in with the reviewers here.

Round 2

Reviewer 2 Report

The revisions are satisfactory. Some of the content in the methods can be put into the results. The authors needs to polish the language before the paper can be accepted.

Author Response

Reply

Many thanks to everybody who participated in the evaluation process. We have made the necessary adjustments to the revised text in order to adhere to the reviewer's recommendation that the techniques be included in the conclusions. In addition to that, we have improved the content by making the language more clear. Please check again.

Reviewer 4 Report

Although the author has made many modifications, I think it is still necessary to improve the research area map, theoretical framework and technical route.

Author Response

Response: Based on the reviewers' suggestions, we have made the major improvements to the study area map, the theoretical framework, and the technical route to the maximum feasible. The following is a summary of the updates:

    China's share of the world's resources per person is one twelfth of what it is in the United States and one ninth of what it is in the United Kingdom. This is in comparison to other developed countries, such as the United States and the United Kingdom. Yet, because of the continued expansion of China's economy, the demand of Chinese citi-zens for raw water of a high quality has gradually increased. This is a problem for China's water supply. Because of this, there is now a disparity between the needs of Chinese residents and the availability of actual resources [16]. The wide mode of eco-nomic development that has been created over a long period of time has led to a poor efficiency of resource consumption. This is the result of the low efficiency of resource usage. This unjustifiable manner of development is also reflected in China's approach to the development and usage of its leisure agricultural resources. The wide style of economic development in China has resulted in a poor level of resource utilization ef-ficiency in rural areas, despite the fact that these rural areas contain a wealth of land and natural resources. As can be observed in Figure 2, the utilization efficiency of lei-sure agriculture in China is much lower when compared to that of developed countries such as Europe and the United States.

Round 3

Reviewer 4 Report

Although the author has improved the article, it has not impressed me in terms of innovation or article structure.